# Epigenetic Landscape Is Largely Shaped by Diversiform Transposons in *Aegilops tauschii*

**DOI:** 10.3390/ijms24119349

**Published:** 2023-05-27

**Authors:** Chuizheng Kong, Guangyao Zhao, Lifeng Gao, Xiuying Kong, Daowen Wang, Xu Liu, Jizeng Jia

**Affiliations:** 1National Key Facility for Crop Gene Resources and Genetic Improvement, Institute of Crop Sciences, Chinese Academy of Agricultural Sciences, Beijing 100081, China; czkong@126.com (C.K.); zhaoguangyao@caas.cn (G.Z.); gaolifeng@caas.cn (L.G.); kongxiuying@caas.cn (X.K.); 2State Key Laboratory of Wheat and Maize Crop Science, College of Agronomy, Henan Agricultural University, Zhengzhou 450002, China; dwwang@henau.edu.cn

**Keywords:** chromatin state, histone modification, chromatin accessibility, wheat, transposons

## Abstract

Transposons (TEs) account for more than 80% of the wheat genome, the highest among all known crop species. They play an important role in shaping the elaborate genomic landscape, which is the key to the speciation of wheat. In this study, we analyzed the association between TEs, chromatin states, and chromatin accessibility in *Aegilops tauschii*, the D genome donor of bread wheat. We found that TEs contributed to the complex but orderly epigenetic landscape as chromatin states showed diverse distributions on TEs of different orders or superfamilies. TEs also contributed to the chromatin state and openness of potential regulatory elements, affecting the expression of TE-related genes. Some TE superfamilies, such as hAT-Ac, carry active/open chromatin regions. In addition, the histone mark H3K9ac was found to be associated with the accessibility shaped by TEs. These results suggest the role of diversiform TEs in shaping the epigenetic landscape and in gene expression regulation in *Aegilops tauschii*. This has positive implications for understanding the transposon roles in *Aegilops tauschii* or the wheat D genome.

## 1. Introduction

Common wheat is the most widely distributed staple crop in the world [1]. It is an allohexaploid composed of three subgenomes: A, B, and D. The success of common wheat is attributed primarily to the three subgenomes and their coordination, especially post the addition of the D genome [2,3,4]. *Aegilops tauschii* is the ancestor of the D genome of hexaploid wheat [5]. The addition of the D genome plays an important role in enhancing environmental adaptability, stress resistance, and grain quality [6,7], and this event of polyploidization is relatively recent [8]. The study of the origin or evolution of the wheat D genome is routinely studied using *Ae. tauschii*. Transposons (TEs) are important markers of speciation and the main reason for genome amplification [9,10]. The vast majority of sequences in the genome of *Ae. tauschii* are repetitive sequences, with TEs accounting for 85.9% of the genome, higher than any other known crop species [6]. A growing number of studies have shown that TEs are not junk sequences, but instead, contribute to the unique resilience and environmental adaptability of species [11,12,13]. For example, in plants, the TEs of wheat contribute to the stability and compatibility of the subgenomes, as well as the adaptability to various environments [14,15]. In animals, the TEs of Antarctic krill account for its huge genome, enormous biomass, and adaptation to the cold and highly seasonal Antarctic environments [16].

Studies involving epigenome profiling of hexaploid wheat [17,18,19] mainly focus on chromatin characteristics and expression regulation of homoeologous genes and prediction of regulatory elements and provide new insights into cis and trans regulation. However, relatively few studies have reported the types of epigenetic modification and chromatin openness produced by TEs and their functions. TEs play a crucial role in the expansion, evolution, and stability of the wheat genome [20,21,22], as well as the evolution of the epigenome [23,24]. TEs can be classified into retrotransposons (Class I elements) and transposons (Class II elements) on the basis of the transposition mechanism, each of which can be further classified into different orders and superfamilies based on structure [25]. The subgenome-specific TEs and their DNA methylation were found to mediate higher-order chromatin structural interactions between subgenomes [26].

The chromatin states and accessibility are dynamic during wheat polyploidization. TEs in the ancestral species made different contributions to hexaploid wheat. The convergent and divergent regulation of wheat subgenomes is attributed to ancient TE expansions before the divergence of the diploid ancestors and the emerging DNA expansions following the subgenome divergence, respectively, affecting the plasticity of polyploid wheat regulation [27]. Changes in the histone modifications and accessibility of TEs during polyploidization can affect the stability and function of the wheat genome. For example, changes at the level of H3K27me2 are correlated with the degree of silencing of DTC transposons and the increased modification led to genome stability and genetic recombination [28]. The reduced chromatin accessibility of the 3L chromosome arm of *Ae. tauschii* in hexaploid wheat resulted in an overall decrease in gene expression in this chromosome arm [29]. In addition, nucleolar dominance and centromere shifts after wheat polyploidization are closely related to changes in epigenetic modification or chromatin accessibility of the constitutive TEs [30,31].

Histone modifications and chromatin accessibility are important components of the epigenetic landscape. Studies involving polyploid plants have shown that parental inheritance is dominant in the differentiation of polyploid subgenomes [32]. Wheat genetics/epigenetics are likely to be largely determined by the diploid ancestor and their divergence involving TEs. Epigenomic studies of wheat ancestral species have been reported in *Triticum urartu* (2n = 14; AA), the A-genome progenitor [33], where TEs are found significantly and continuously shaping regulatory networks related to wheat genome evolution and adaptation. However, the epigenetic characteristics, especially those of TEs, of *Ae. tauschii* (2n = 14; DD) are still unclear. The *Ae. tauschii* epigenome exploring TEs will shed light on studies involving the wheat D epi-subgenome and provide important insights into the relationship between the formation of the epigenetic landscape and the TEs in wheat.

Here, we used data from chromatin immunoprecipitation and sequencing (ChIP-seq) and micrococcal nuclease digestion and sequencing (MNase-seq) to profile the histone modification and chromatin accessibility maps of *Ae. tauschii* AL8/78 and investigated the chromatin states and open regions on TEs. TE-associated chromatin states accounted for 92.0% of the genome, with a strongly diverse state composition of TE superfamilies. In addition, TE superfamilies were found related to the formation of chromatin states of genes and regulatory elements, which has a positive significance for gene expression regulation.

## 2. Results

### 2.1. Chromatin State Profiling of Ae. tauschii

Based on the ‘epigenetic code’ hypothesis [34], we used a variety of histone modification marks to capture different types of chromatin modifications in *Ae. tauschii* (Appendix A). These marks represent a wide range of genomic elements and can comprehensively reflect the epigenetic landscape of the wheat D genome. The effectiveness of these commercial histone antibodies has been confirmed in our previous study (article under revision). Different biological replicates have been identified to have peak regions (highly enriched mark regions) with good repeatability.

According to the combinatorial pattern of these marks, we used the multivariate hidden Markov model (HMM) [35] to determine the chromatin state of *Ae. tauschii*, dividing the AL8/78 genome into 15 chromatin states (Figure 1a). The different states showed good discrimination and represented different biological units in the genome (Figure 1b,c). We further annotated the corresponding regions of all the chromatin states and classified the chromatin states into TE regions, regulatory elements, genes, centromeres, and other modified states (Table 1).

Chromatin states 1–9, which are TE-associated states that maintain genome stability, mainly enriched in the heterochromatin regions and accounted for 92.0% of the genome (Figure 1a,b, Table 1). These states included constitutive heterochromatin type (mainly modified by H3K27me2 and H3K9me2/3) and facultative heterochromatin type (mainly modified by H3K27me3) (Figure 1a,b). Chromatin states 10–13 were mostly distributed near the genes, and these regions are enriched in flanking TSSs, active/inhibitory TSSs, and TESs (Figure 1b,d). Chromatin state 14 was found mainly in the intragenic regions, enriched in H3K36me3, H2Bub, and H3K4me1, the modifications of gene bodies or modifications accompanying transcription (Figure 1a,b, Table 1). Chromatin state 15 was mostly enriched in centromere-specific histone H3 variant (CENH3), representing the major state of centromeric regions (Figure 1a,b).

### 2.2. Chromatin State Signatures on TE Orders

We first calculated the enrichment of all chromatin states in TE orders (Figure 2a). We found that LTRs and DNA transposons, accounting for the majority (58.24% and 19.19%) of the genome, were mainly distributed in states 1–9 and 15 and contributed to the main heterochromatin regions. State-15-covered regions, the centromeric regions defined by CENH3, were mainly composed of LTR-type TEs, which was consistent with the composition of the repetitive sequence of the centromeres, including *CRWs* and *Quintas*, the two types of LTR [36]. A substantial portion of LINEs was distributed in chromatin states 9–11, which were closely related to the enrichment of H3K27me3 (Figure 2a). Chromatin state 9 was enriched in almost all TE orders and was associated with the presence of H3K18ac. In addition, the TE-enriched chromatin states of different orders differed, even though these orders belong to the same class. For example, LTR, LINE, and SINE all belong to Class I; however, their chromatin states differed clearly (Figure 2a).

### 2.3. Chromatin State Signatures on TE Superfamilies

We further divided TEs into superfamilies, the classification under TE orders, and calculated the fold enrichment of 15 chromatin states on TE superfamilies whose repeat numbers were greater than 100 (see Section 4). The relative enrichment distribution of chromatin states across different superfamilies shows a ‘mosaic’ pattern (Figure 2b), indicating that the differentiation of TE chromatin states was also reflected by the TE superfamily. We found a high degree of correspondence between certain TE superfamilies and chromatin states (Appendix A). For example, by calculating the absolute fold enrichment of TEs in states, we found that LINE/R1 and SINE/L1 were highly enriched in specific chromatin states: LINE/R1 was enriched 33.4-fold in state 11, and SINE/L1 was enriched 23.1-fold in state 9, relative to random expectation (Appendix A). These two states were also mainly found in these two TE superfamilies (Figure 2b).

We also found that TE superfamilies may contribute to the chromatin states of regulatory elements and genes (Figure 2b and Appendix A). For example, most DNA/Helitron, DNA/hAT-Tag1, and LINE/RTE participated in the chromatin state of genes (state 14); LINE/Jockey was mainly enriched in the activated chromatin state (i.e., state 13), and DNA/Tc1 was found to be mainly enriched in the inhibited chromatin state (i.e., state 10) (Figure 2b and Appendix A). TEs also exhibited unmodified chromatin regions. For example, chromatin state 5 represented the unmodified histone region, accounting for 37.3% of the genome, and some TE superfamilies such as DNA/Sola and LINE/CRE were found to be extensively enriched in chromatin state 5 (Figure 2b, Table 1).

### 2.4. Chromatin Accessibility and Transcription of Chromatin States

Chromatin accessibility or openness is another important epigenetic characteristic besides the chromatin state [37]. The openness of chromatin is crucial to the regulation of gene expression. MNase-seq is a method to study chromatin openness based on DNA sensibility to MNase digestion [38]. The MNase hypersensitive sites (MNase HSs or HSs) usually indicate *cis* regulatory elements or the binding sites of *trans* factors, that is, the open chromatin regions. We herein identified the MNase HSs in *Ae. tauschii* AL8/78 genome using the MNase-seq (see Section 4).

The MNase HSs identified were validated in our previous study and highly recaptured the open signals detected by ATAC-seq and DNase-seq (article under revision). We analyzed the openness of the chromatin states and the expression of state-related genes and found that states 11–13 showed the highest openness, while the openness of states for gene bodies and TEs was basically low (Figure 3a). The genes related to states 12–14 were strongly expressed, while the genes related to state 11 (Bivalent state) and state 10 (H3K27me3 polycomb) were hardly expressed (Figure 3b). However, the openness of state 11 was much higher than that of state 10 (Figure 3a,b), suggesting the bivalent state 11 is open but suppressive. In addition, states 1 and 2 were accompanied by slight gene expression, which was related to additional state-related genes compared with other TE-associated states (Figure 3b and Appendix A).

### 2.5. Open Chromatin Regions on TEs

Although the MNase HSs were highly enriched near AL8/78 genes, especially in gene promoter regions (Appendix A), 18.6% (3006/16,150) of the highly credible HSs (Bayes factor criterion, strong signals) were found on TEs, indicating the distribution of a small number of highly open regions on TEs. To analyze the contributions of different TE superfamilies to open chromatin regions, we calculated the enrichment of 3006 highly credible HSs on TEs (abbreviated to TE-HSs) in TE superfamilies and found that TE-HSs were mainly distributed in Unknown, LTR/Gypsy, LTR/Copia, and LINE/L1, accounting for 81.8% of all TE-HSs (Figure 4a). Based on the relative enrichment of TE-HSs on TE superfamilies (see Section 4), TE-HSs were highly enriched on DNA/hAT-Ac, with the enrichment reaching 52.8-fold compared to the expected value (Figure 4b), suggesting this type of TE is more likely to form open regions in *Ae. tauschii* genome.

**Figure 3 ijms-24-09349-f003:**
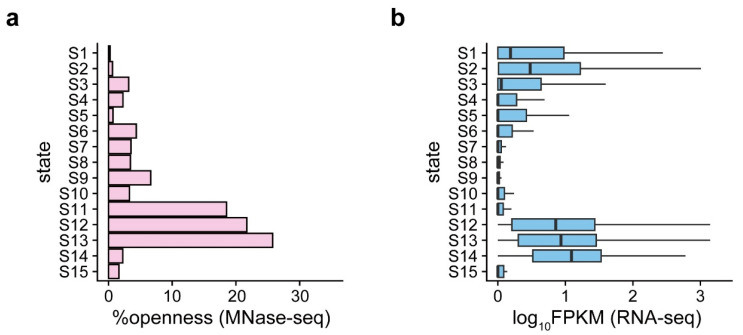
Chromatin openness and expression of related genes in 15 chromatin states. (**a**) The bar chart represents the proportion of the open regions (identified with MNase-seq) in the chromatin states. (**b**) The boxplot shows the expression level (identified with RNA-seq) of related genes in chromatin states.

We found that 57.9% of the hAT-Ac elements with HSs were distributed within the 5 kb gene promoters and 76.3% within the 10 kb gene promoters, substantially higher than the expected ratios of 22.7% and 33.0% for the total hAT-Ac, indicating that hAT-Ac was more likely to contribute to the openness in the promoters. The sequence set of HSs on these hAT-Ac was used to predict the possible transcription factor-binding motifs and found that the three motifs significantly enriched on these HSs (Figure 5, E-value = 5.5 × 10^−20^). These motifs were found enriched in GCC/GGC, consistent with the characteristics of open regions on promoters reported in wheat [19]. We found that two of the motifs were highly similar to the transcription factor-binding motifs of the ERF/DREB family in *Arabidopsis* [39,40] (see Section 4) (Appendix A, *p*-value = 6.58 × 10^−4^).

### 2.6. Impacts of TE Chromatin States on Openness and Gene Expression

To determine the association between TE-associated chromatin states, open regions of TEs, and TE-related gene expression, we classified the TE-HSs based on their distance from the genes. TE-HSs within 1 kb upstream and 1 kb downstream TSSs were defined as proximal TE-HSs. TE-HSs beyond 3 kb upstream of TSSs were defined as distal TE-HSs. While 50.2% (8111/16150) of the HSs were distal, 81.2% (2441/3006) of the TE-HSs were distributed distally. We found that distal TE-HSs were predominant in Unknown, LTR/Gypsy, LTR/Copia, and LINE/L1, while proximal TE-HSs were mostly present in LINE/L1 and DNA/HAT-Ac, followed by LTR/Gypsy and LTR/Copia (Table 2). This indicates that LINE/L1 and DNA/hAT-Ac contribute the most to the proximal TE-HSs, and the enrichment of HSs on hAT-Ac elements is more likely to occur in the gene proximal promoters (Figure 4b, Table 2).

GO enrichment analysis showed that the genes with proximal TE-HSs were mostly enriched in binding and catalytic activity functions (Appendix A). We then classified the genes based on the presence of TEs in the proximal promoters and the proximal HSs on TEs and found that the TE-associated genes had lower expression levels than those without TEs (Figure 6a, Wilcoxon test), indicating that the TE presence is not conducive to gene expression. However, the expression of genes with proximal TE-HSs was significantly higher than that of genes without HSs (with or without TEs) (Figure 6a, Wilcoxon test), indicating that TE insertions with HSs actually play a role in gene expression activation compared with the absence of HSs in gene promoters. These results indicate that the formation of open chromatin regions on TEs can affect the expression of adjacent genes.

The distal HSs usually indicate remote regulatory elements, such as enhancers, which act in long-distance gene regulation [41]. We focused on the distal LTR/Gypsy with the highest proportion of TEs and the highest content of distal HSs (Table 2). First, the distal LTR/Gypsy was divided into two types, with and without HSs, based on overlap with the distal HSs. LTR/Gypsy with HSs enriched additional states 12 and 13 compared with LTR/Gypsy without HSs, although LTR/Gypsy with or without distal HSs both enriched constituent heterochromatin states dominated by state 7 (Figure 6b). These distal HSs were located far away from TSSs (Table 2), so states 12 and 13 were involved in the activation of some distal *cis* regulatory elements. Together, these results suggest a close correlation between the activated chromatin states and the formation of open regions on TEs.

We further profiled the distributions of H3K9ac and H3K9m2, which are representative marks of state 12/13 and state 7, respectively, around distal LTR/Gypsy (Figure 6c). Compared with LTR/Gypsy without HSs or genomic random regions, LTR/Gypsy carrying HSs were further enriched in H3K9ac, especially at both ends of LTR/Gypsy (Figure 6c). The modification of H3K9me2 in LTR/Gypsy was slightly enriched, but compared to the input control, the H3K9me2 enrichment was not significant (Figure 6c and Appendix A). As a control, H3K27me3 in LTR/Gypsy was largely unaffected by the presence of HSs (Appendix A). We profiled the enrichment of histone marks around the HSs on distal LTR/Gypsy and found that H3K9ac was significantly higher in the HSs and their flanking regions than in LTR/Gypsy shuffled regions (Figure 6d), indicating that the HSs were mainly located at the H3K9ac modified regions on distal LTR/Gypsy. These results indicate that some active histone modifications such as H3K9ac may be closely related to the formation of activated chromatin states and open chromatin regions of TEs.

## 3. Discussion

Wheat provides nearly 20% of the total dietary calories and proteins worldwide and feeds almost 35% of the world’s population [42,43]. The availability of the wheat D genome has led to the worldwide expansion of wheat from the Middle East. The D genome has enhanced the adaptability of wheat to the global environment and increased its resistance to biotic and abiotic stresses [3,7,8]. Currently, wheat production faces great challenges due to environmental changes, such as extreme climate and drought [44,45]. The identification of the factors and possible mechanisms driving the response of the D genome to such changes in wheat breeding and cultivation is crucial to ensure sustainable wheat production and food security. TEs and epigenetics (such as chromatin state and chromatin openness) are important factors contributing to genome stability, regulation of gene expression, and the evolution of environmental adaptation in wheat [15,22,26,46]. However, the role of various TEs in shaping the epigenetic landscape of the wheat genome is unclear. The study of *Ae. tauschii* genome revealed that TEs are significantly associated with other genomic features such as gene and pseudogene density, miRNA levels, and gene expression and recombination, highlighting the multiple effects of TEs on *Ae. tauschii* genomic landscape [6]. In this study, we focused on the chromatin state of TEs in the hierarchy of TE orders and superfamilies. The TEs accounting for 85.9% of the *Ae. tauschii* genome contributed to approximately 92.0% of the chromatin states, which is very similar to that in maize, but much higher than that in *Arabidopsis* and rice [47] (Appendix A). TEs and chromatin states are highly correlated and may have an important role in shaping the epigenomic landscape of *Ae. tauschii*. Our results further highlight the contribution of TEs to *Ae. tauschii* as well as to the wheat D genome.

Furthermore, the selection of an appropriate method plays a key role in enhancing the study outcomes. We constructed a comprehensive epigenetic landscape of *Ae. tauschii* based on histone modification and chromatin accessibility maps, which played a crucial role in determining the epigenetics of the wheat D genome. The marks we used include histone methylations, acetylations, ubiquitination, and histone variant, which are widely used in investigating plants such as *Arabidopsis* [48], rice [49], and wheat [19]. According to the hypothesis of ‘epigenetic code’ and ‘histone code’ [34,50], the potential of gene expression in a specific cell type can be determined based on the state of chromatin modification during the early stages of differentiation. The combination of chromatin modifications determines the switch in gene transcriptional status of specific developmental or differentiation stages [41,51]. The combination of chromatin modifications or histone modifications is also referred to as the chromatin state [48]. Chromatin states 1–9 accounted for 92.0% of the genome, whereas chromatin states 10–13 constituted 5.9% of the genome, and chromatin state 14 1.7% of the genome (Table 1). These proportions were similar to those of TEs, regulatory elements, and genes in the *Ae. tauschii* AL8/78 genome [6], which reinforces our study findings involving chromatin state learning and genomic annotation.

Chromatin openness and gene expression in wheat are related to the chromatin state [19]. Our results support previous research findings. However, state 11 exhibited a different pattern (Figure 3), suggesting a bivalent chromatin state with active and repressive marks. The active marks mainly included H3K4me3 and H3K18ac, and the repressive mark was mainly H3K27me3. State 11 was open but not accompanied by the expression of related genes, suggesting that opening is necessary but insufficient for expression. Previous studies have reported that the binding of transcription inhibitors or repressive modification may interfere with gene expression [52,53]. In addition, in this study, some TEs located near the genes may also contribute to the association between TE-related chromatin states and gene expression. For example, genes related to states 1 and 2 were found to be highly expressed in our study and associated with additional related genes (Figure 3b and Appendix A).

The study investigating the chromatin state is the earliest and most comprehensive involving mammals, such as *Homo sapiens* and *Mus musculus* in the ENCODE project [54,55]. Active histone modification may differ in plants and mammals, while the major histone modifications of TE may be relatively conserved [47]. Studies involving *Arabidopsis* and rice revealed that the chromatin states associated with H3K9me2 and DNA methylation were closely related to TEs [48,56,57], which was consistent with a subsequent study involving wheat [19]. In this study, H3K9me2 and H3K27me2/3 were found to be strongly associated with TEs in *Ae. tauschii* (Figure 1a). Furthermore, we found that the TE-related H3K9me3 in mammals was also closely associated with TEs in *Ae. tauschii*, with distribution patterns different from those of H3K9me2 (Figure 1a). Compared with the conserved TE changes, the number of TEs and their classification vary across plants and animals and also between different plant species [10,58]. Our results indicate that the TE chromatin states are altered at the level of order and superfamily. Diverse chromatin states were enriched in different TE superfamilies (Figure 2b). The differentiation in structure and core elements of TE superfamilies might play a role in their differentiation in chromatin states. Furthermore, the superfamilies under different classes may express similar chromatin states, such as DNA/Sola and LINE/CRE, which may be related to shared elements of TEs, as well as the location of TEs.

A recent study involving plants suggests that open chromatin regions can be created by TEs and translocated via TE proliferation [59]. TE expansion is a general feature that contributes to open chromatin regions. We found that a few TE superfamilies tend to carry open regions, such as the hAT-Ac superfamily, which was strongly enriched in MNase HSs, especially in proximal gene promoters (Figure 4b, Table 2). This was not reported in the previous study; however, we found that TEs of hATs indeed enrich open chromatin regions in a large number of plant species [59]. DNA/hAT-Ac is a type of autonomous ‘cut-and-paste’ TE, which is widely reported in maize [60], snapdragon [61], and fruit flies [62], and is supposed to transpose and induce gene activation. The presence of transcription factor-binding motifs on hAT-Ac and the spatial location of hAT-Ac after transposition may induce the formation of open regions in hAT-Ac. Furthermore, the open regions that regulate the expression of neighboring genes and improve the environmental adaptability of *Ae. tauschii* may also contribute to the enrichment of such HSs. In *Arabidopsis*, the ERF/DREB family acted as transcriptional activators and bound to the GCC-box pathogenesis-related promoter element during the regulation of gene expression by stress factors and by components of stress signal transduction pathways [63].

Increased nuclease sensitivity and H3K9ac or H3K27ac are typical characteristics of regulatory elements, including promoters and enhancers, which were reported in maize and wheat [19,64]. In this study, H3K9ac and H3K27ac were the major marks of the chromatin state 12/13. LTR/Gypsy with distal HSs carried a higher number of chromatin states 12/13 and H3K9ac modification was increased significantly at distal HSs (Figure 6b,d), implying that these distal TEs may carry activated enhancer elements in the HSs. Further, the open signals on TEs in the proximal promoters were correlated with gene expression (Figure 6a), consistent with the gene expression induced by open promoters in rice [65]. Interestingly, our study revealed that genes with TE neighbors carrying HSs showed higher levels of expression compared with genes carrying TE neighbors without HSs (Figure 6a), indicating a positive significance of TE opening near genes.

Overall, our results demonstrate that TEs of different orders or superfamilies are differentiated on the enrichment of different chromatin states in the *Ae. tauschii* genome. A few TE superfamilies were found to be readily enriched in open chromatin regions, suggesting that diverse TEs contribute to the shaping of the epigenetic landscape of *Ae. tauschii*. Our results contribute to our understanding of the relationship between TE abundance and epigenetic modifications, as well as the role of TEs in the formation of regulatory elements in the wheat D genome. The results provide new insights into the mechanisms of TEs shaping the epigenetic landscape.

## 4. Materials and Methods

### 4.1. Plant Materials and Growth Conditions

*Ae. tauschii* accession AL8/78 was used for this study. AL8/78 is an accession collected by V. Jaaska near the Hrazdan River, Jerevan, Armenia. The information on AL8/78 can be found on the website (http://aegilops.wheat.ucdavis.edu/ATGSP/, accessed on 2 March 2022). The accession was previously selected for reference genome sequencing for its genetic proximity to the wheat D genome and as it has been extensively characterized genetically [6,66]. Seeds were pre-sterilized with 1% hydrogen peroxide and germinated in Petri dishes, and seedlings in consistent growth status were transferred to nutrient solution for further cultivation until 21 days after germination when plants are at the three-leaf stage. The environmental conditions for the light incubator during growth were set to 20 °C/18 °C (day/night) and 16 h/8 h (light/dark). Leaves were harvested, frozen in liquid nitrogen, and stored at −80 °C. Samples used for ChIP-seq need to be pretreated in a formaldehyde-containing fixation buffer before freezing and preservation.

### 4.2. ChIP-Seq and MNase-Seq

We referred to a previous experimental protocol used for Chip-seq and used IgG as a negative control [19]. Samples from >10 plants were harvested for each experiment, and 10–30 ng captured DNA as well as uncaptured input DNA were used to construct ChIP-seq libraries. The data were sequenced using the Illumina platform HiSeq6000 system, yielding 150-bp paired-end reads. A total of 100 million reads (~30 G) were generated for each biological replicate, and the two biological replicates were conducted. We referred to the method used in the maize studies for MNase-seq and made moderate modifications [38,67]. We used 1% agarose gel to extract the 100–200 bp DNA fragments and purified the DNA using the Qiaex II gel extraction kit (Qiaex, Hilden, Germany). Approximately 1 μg DNA was used to construct the MNase-seq library. The high-throughput sequencing platform and sequencing method were the same as ChIP-seq.

### 4.3. Learning of Chromatin States in Ae. tauschii

For ChIP-seq, we evaluated the data quality using FastQC v0.11.5 software [68] and cleaned the data with Trimmomatic v0.3.6 software [69], including removing sequencing adaptors, trimming reads with 5′ and 3′ end quality scores lower than 5 (Phred + 33) and discarding trimmed reads with length < 20 bp. Using Bowtie2 v2.3.4 software [70] with ‘--very-sensitive’ and ‘--end-to-end’ parameters, the clean data were aligned to the AL8/78 genome reference [6]. The alignments with MAPQ < 20 were discarded to obtain unique reads mapping. Duplicate reads resulting from PCR amplification were removed using the markdup function in the Samtools software (https://sourceforge.net/projects/samtools/, accessed on 17 April 2023) [71]. The bam files of histone marks beyond H3 were used as input files to learn the chromatin states of AL8/78 using the BinarizeBam and LearnModel functions in ChromHMM v1.19 software [35]. The maximum likelihood estimation was used to determine the number of chromatin states.

### 4.4. Identification and Classification of Open Chromatin Regions

We used the same quality control method as the ChIP-seq and aligned the MNase-seq clean data to the AL8/78 genome reference using Bowtie2 v2.3.4 software with parameters ‘--no-mixed --no-discordant --no-unal --dovetail’. Alignments with MAPQ < 20 as well as duplicate reads resulting from PCR amplification were identically removed. The depth of read coverage in 10 bp bins was calculated using the coverage function in the Bedtools software [72] (https://www.encodeproject.org/software/bedtools/, accessed on 17 April 2023). The heavy-digestion coverage depths were subtracted from the light-digestion ones, and differential nuclease sensitivity sites (DNSs) were obtained. The positive DNSs were defined as MNase hypersensitive sites (HSs), and the negative DNSs were defined as MNase hyper-resistant sites (HRs). Regions with highly credible HSs were further defined using the Bayes factor criterion. Highly credible MNase HSs located within 1 kb upstream and 1 kb downstream of TSSs were defined as proximal HSs, and highly credible MNase HSs within 3 kb upstream of TSSs were defined as distal HSs.

### 4.5. Identification of Chromatin States on TE Orders and Superfamilies

TEs were classified according to order or superfamily, and we used the OverlapEnrichment function in ChromHMM v1.19 software to compute the fold enrichment of each state on TEs of the superfamily. The intersection in the BEDtools software was used to obtain the TEs or HSs overlapped with the chromatin states. The computeMatrix and plotProfile in the deepTools software [73] (https://pypi.org/project/deepTools/, accessed on 17 April 2023). were used to calculate and plot histone modification enrichment around the TEs.

### 4.6. Calculation of State and HS Enrichments on TEs

The fold enrichment calculation is as follows: let A be the number of bases in the state, B be the number of bases in the TEs, C be the number of bases in the state and the TEs, D be the number of bases in the genome, E be the count of HSs in the genome, F be the length of TEs, G be the count of HSs in the TEs, and H be the length of the genome. The state fold enrichment and HS fold enrichment are then defined as (C/A)/(B/D) and (G/E)/(F/H), respectively.

### 4.7. Prediction and Similarity Quantitation of Motifs

The MEME-ChIP website [74] (https://meme-suite.org/meme/tools/meme-chip, accessed on 25 July 2022) was used for motif prediction, and the JASPAR (NON-REDUNDANT) CORE plants database [75] was used for known motif searching. Tomtom website [76] (https://meme-suite.org/meme/tools/tomtom, accessed on 25 July 2022) was used to predict the similarity to the known motifs. Uniprot Website [77] (https://www.uniprot.org/, accessed on 25 July 2022) was used to view the function of the predicted transcription factors.

### 4.8. Identification of TE-Associated Chromatin States in Arabidopsis, Rice, and Maize

The chromatin state data used in this study for *Arabidopsis thaliana*, *Oryza sativa*, and *Zea mays* was obtained from the Plant Chromatin State Database [47] (PCSD, http://systemsbiology.cau.edu.cn/chromstates, accessed on 13 May 2023). We identified the TE-associated chromatin states of the three species based on fold enrichment and preferential location of the TEs. These data were then compared with our data in *Ae. tauschii* (Appendix A).

## 5. Conclusions

In conclusion, our study reveals that TEs in the *Ae. tauschii* genome are associated with most of the chromatin states, with a diverse state distribution across TE orders and superfamilies. TEs were associated with chromatin accessibility and related gene expression. Furthermore, hAT-Ac was enriched substantially in open regions at higher levels than the expected value and was concentrated in the proximal gene promoters. We identified motifs highly similar to the transcription factor-binding motifs of the *Arabidopsis* ERF/DREB family. The enrichment of active histone modifications such as H3K9ac and activated chromatin states on gene-distal TEs were highly correlated with the formation of open regions. The results provide insight into the role of TEs in shaping the epigenetic landscape and highlight the importance of TEs in *Ae. tauschii* or wheat D genome.

## Figures and Tables

**Figure 1 ijms-24-09349-f001:**
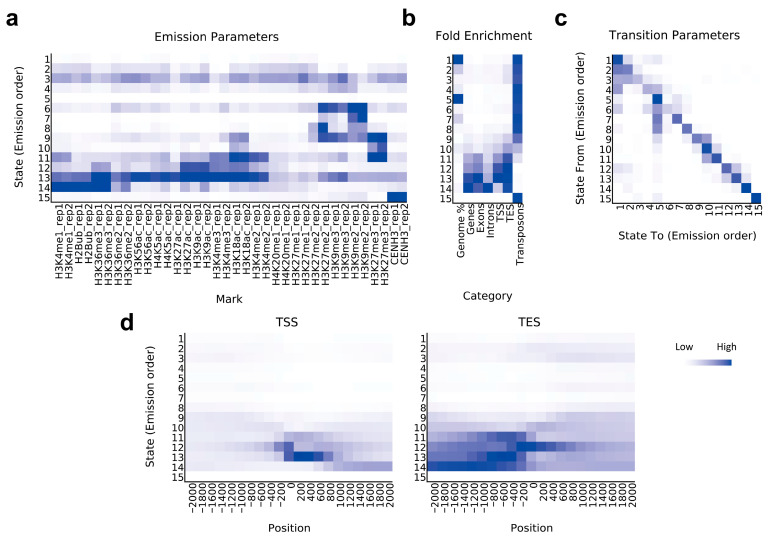
Determination and distribution of chromatin states in *Ae. tauschii*. (**a**) Chromatin states are determined with a multivariate hidden Markov model. The heatmap presents the emission parameters based on genome-wide combinations of histone marks. (**b**) The overlap enrichment of chromatin states in genomic features. (**c**) Pair-wise comparison of 15 chromatin states. (**d**) The neighborhood enrichment of chromatin states at transcription start sites (TSSs) and transcription end sites (TESs). For all heatmaps, the darker blue color corresponds to a greater probability or fold enrichment of observing the mark in the corresponding area.

**Figure 2 ijms-24-09349-f002:**
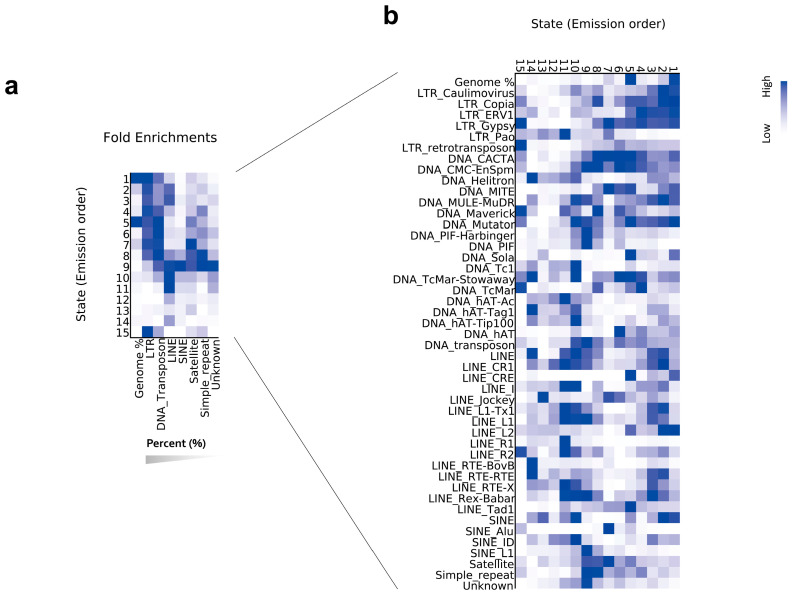
Chromatin signatures on TEs. (**a**,**b**) The heatmaps present the overlap enrichment of chromatin states in TE orders (**a**) and in TE superfamilies (**b**). TE orders are presented in descending order according to their proportions in the genome. The darker blue color corresponds to a greater fold enrichment of observing the mark in the TEs.

**Figure 4 ijms-24-09349-f004:**
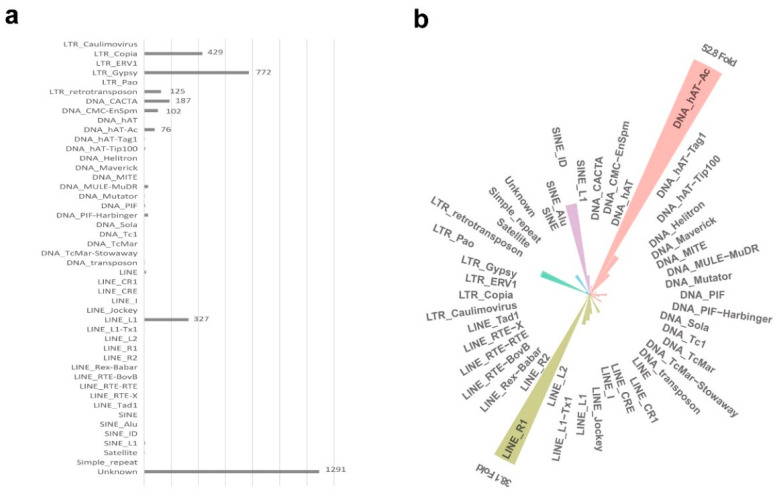
Enrichment of MNase HSs on TEs. (**a**) The bar chart represents the numbers of highly credible HSs on TEs (TE-HSs) detected in TE superfamilies. (**b**) The circular diagram represents the relative fold enrichment of TE-HSs.

**Figure 5 ijms-24-09349-f005:**
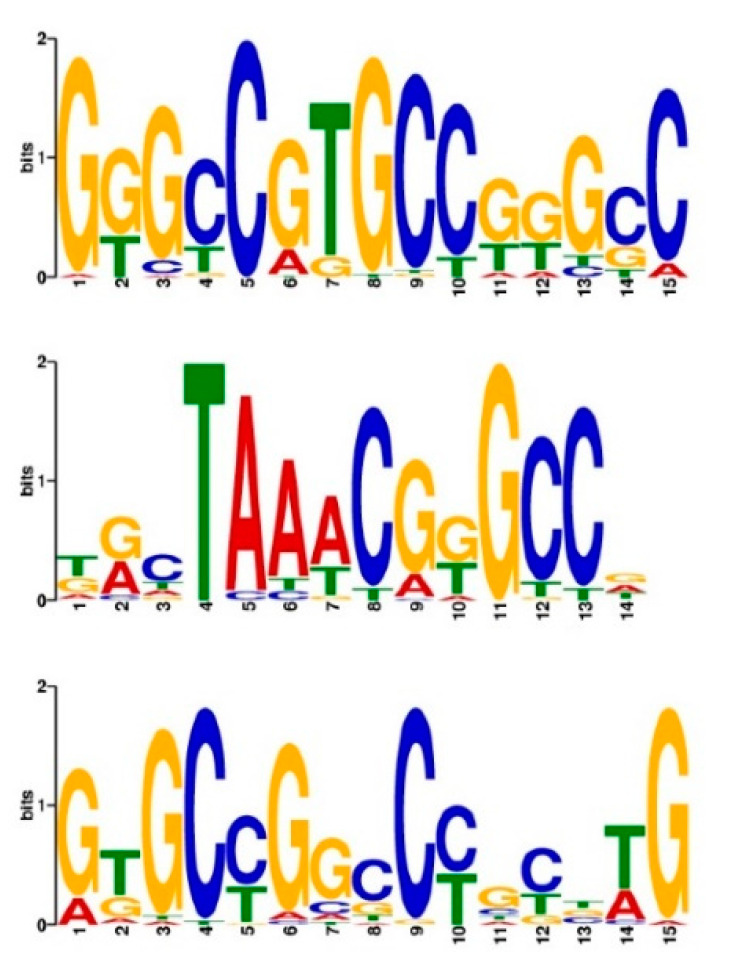
DNA motifs predicted in the MNase HSs of hAT-Ac superfamily.

**Figure 6 ijms-24-09349-f006:**
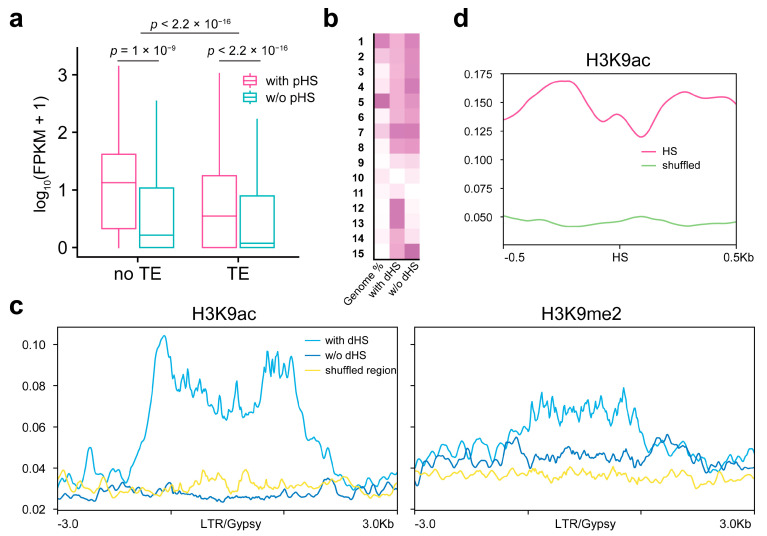
The relationship between TE chromatin states, openness, and gene expression. (**a**) The boxplot shows the effect of TEs on gene expression and the effect of TEs with proximal HS (pHS) on gene expression. (**b**) The heatmap presents the overlap enrichment of chromatin states in LTR/Gypsy with or without distal HS (dHS). Colors from light to dark represent the degree. (**c**) The profiles show H3K9ac and H3K9me2 distributions in LTR/Gypsy with or without dHS. Shuffled genomic regions represent control. (**d**) The profile shows H3K9ac distribution around HSs on distal LTR/Gypsy. Shuffled regions on LTR/Gypsy are used as control.

**Table 1 ijms-24-09349-t001:** Genomic annotation of the chromatin states in *Ae. tauschii*.

State	Annotation	Length (bp)	Percent (%)	Summary (%)
S1	TE region 1	1,227,693,800	30.65	S1–S9 (92.02)
S2	TE region 2	314,684,600	7.86	
S3	TE region 3	72,102,000	1.80	
S4	TE region 4	136,383,200	3.40	
S5	TE region 5 (Unmarked)	1,494,160,200	37.30	
S6	TE region 6	74,273,600	1.85	
S7	TE region 7	281,086,000	7.02	
S8	TE region 8	60,162,800	1.50	
S9	TE region 9 (H3K18ac-assosciated)	25,231,000	0.63	
S10	H3K27me3 Polycomb	125,719,600	3.14	S10–S13 (5.91)
S11	Bivalent state	53,689,600	1.34	
S12	FLanking TSS	26,332,400	0.66	
S13	Active TSS	31,050,800	0.78	
S14	Intragenic region	66,268,600	1.65	1.65
S15	Centromeric region	16,660,000	0.42	0.42
total	#	4,005,498,200	100.00	100.00

**Table 2 ijms-24-09349-t002:** Distribution of distal TE-HSs and proximal TE-HSs in the TE superfamilies (top 15).

Distal TE-HSs		Proximal TE-HSs	
TE Superfamily	Count (Ratio/%)	TE Superfamily	Count (Ratio/%)
Unknown	795 (31.3)	Unknown	160 (57.1)
LTR/Gypsy	701 (27.6)	LINE/L1	34 (12.1)
LTR/Copia	357 (14.1)	DNA/hAT-Ac	29 (10.4)
LINE/L1	216 (8.5)	LTR/Gypsy	19 (6.8)
DNA/CACTA	161 (6.3)	LTR/Copia	8 (2.9)
LTR retrotransposon	114 (4.5)	DNA/CACTA	7 (2.5)
DNA/CMC-EnSpm	79 (3.1)	DNA/CMC-EnSpm	5 (1.8)
DNA/hAT-Ac	33 (1.3)	DNA/PIF-Harbinger	5 (1.8)
DNA/MULE-MuDR	17 (0.7)	DNA/MULE-MuDR	3 (1.1)
DNA/PIF-Harbinger	15 (0.6)	LINE/R1	2 (0.7)
LINE	12 (0.5)	SINE/L1	2 (0.7)
DNA/hAT-Tip100	5 (0.2)	LTR/Pao	1 (0.4)
DNA/PIF	5 (0.2)	LTR/retrotransposon	1 (0.4)
DNA/transposon	4 (0.2)	DNA/hAT-Tag1	1 (0.4)
SINE/L1	4 (0.2)	DNA/Mutator	1 (0.4)

## Data Availability

The raw data of ChIP-seq and MNase-seq used in this study can be available in the National Genomics Data Center (NGDC, https://bigd.big.ac.cn) under project accession number PRJCA012697.

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
