# Peer review of "Epigenetic Landscape Is Largely Shaped by Diversiform Transposons in Aegilops tauschii"

_ijms, 2023, doi:10.3390/ijms24119349_

Round 1
Reviewer 1 Report
The manuscript is elaborated on an interesting topic, but contains methodological and formal shortcomings. In the case of material, it is necessary to clearly specify the source of the material used and characterize it by an unmistakable method (e.g. a code from internationally recognized databases - GRIN, etc.). The presentation of the achieved results is also problematic, because in some places (e. g. line 91, 95, 97, 135) in the "Results" section there are references and swear words that belong to the discussion. On the contrary, the Discussion part is very weak due to the amount of data collected. Comparing your results from one reference, even in a sentence with a general formulation, is really insufficient. Why are the results not discussed with current works on this topic. Especially if the authors use in some graphic appendices a comparison of their results with, for example, Arabidopsis thaliana? Although it is stated in these figures that it is a comparison with A. thalina, the reader is not told where this data comes from, i.e. there is no adequate relevance of documenting the result for their comparison.
The manuscript requires substantial and significant editing of the "Results" and "Discussion" sections. Perhaps it would be appropriate to think about connecting the individual parts. It is through an adequate and critical discussion that authors can prove new knowledge in the given area. At the same time, it would be appropriate to prepare a "Conclusion" section (it doesn't have to be long) with a summary of essential findings.
Author Response
Dear Reviewer,
Thank the reviewer for the comments concerning our manuscript entitled “Epigenetic landscape is largely shaped by diversiform transposons in Aegilops tauschii” (manuscript ID: ijms-2378947). Those insightful and constructive comments are all valuable and very helpful for revising and improving our manuscript, as well as the important guiding significance to our research. We have read the comments carefully and have made corrections which we hope meet with approval. The resubmitted manuscript is under revision mode and revised portions can be seen in our manuscript. The point-to-point responses to your comments are as follows:
The manuscript is elaborated on an interesting topic, but contains methodological and formal shortcomings.
Response: Thanks for the reviewer’s valuable comment. As pointed out by the reviewer, the manuscript indeed contains many methodological and formal shortcomings which cannot be ignored. The following suggestions from the reviewer are very helpful in correcting these shortcomings, improving our manuscript, and better organizing and presenting our topic.
In the case of material, it is necessary to clearly specify the source of the material used and characterize it by an unmistakable method (e.g. a code from internationally recognized databases - GRIN, etc.).
Response: Thank the reviewer for the suggestion. We apologize for our inadequate descriptions of the material. AL8/78 is a Ae. tauschii accession collected by V. Jaaska near the Hrazdan River, Jerevan, Armenia. AL8/78 is the primary accession for the accession code. The information about AL8/78 can be found on the website (http://aegilops.wheat.ucdavis.edu/ATGSP/). We selected AL8/78 as the material used in our study because this accession had been previously selected for reference genome sequencing. AL8/78 was genetically proximal to the wheat D genome and was extensively characterized genetically (Luo et al., 2017, Zhao et al., 2017). We have added the above information to the “Materials and Methods” section and “Acknowledgments” section of the manuscript accordingly.
The presentation of the achieved results is also problematic, because in some places (e. g. line 91, 95, 97, 135) in the "Results" section there are references and swear words that belong to the discussion.
Response: Thank the reviewer for pointing out this mistake. According to the reviewer’s suggestion, we have moved line 91, 95, 97 and 135 in the "Results" section to the “Discussion”. Besides, we also moved line 224 to the “Discussion”.
On the contrary, the Discussion part is very weak due to the amount of data collected. Comparing your results from one reference, even in a sentence with a general formulation, is really insufficient. Why are the results not discussed with current works on this topic.
Response: Thank you for the reviewer’s comments and constructive suggestions. We would like to apologize for our weak discussion, which was not adequately discussed with current works on this topic. Here, we did an in-depth discussion. We rewrote the “Discussion” section and compared our results with existing works on this topic. The main discussions added this time are as follows:
- In the context of wheat as an important staple crop and increasing environmental suitability of wheat, we discuss why tauschii is used, why TE and epigenetic landscape studies are conducted, and compared with studies on TEs in the Ae. tauschii genomic landscape (paragraph 1 in “Discussion”)
- In the context of the hypothesis of the “histone code”, we discuss why and how this method can establish the epigenetic landscape and further discussed the interesting results of TE states and bivalent states (paragraph 2 in “Discussion”)
- Compared with the chromatin state studies in mammals, Arabidopsis, rice, and wheat, we discuss the results of TE-related modifications in tauschii and the differentiation of chromatin states between TE superfamilies (paragraph 3 in “Discussion”)
- We discussed the possible reasons for hAT-Ac enrichment in open areas and compared them with a recent study on the prevalence and chromatin signatures of plant regulatory elements on TEs in plants (paragraph 4 in “Discussion”)
- Compared with the open features of promoter and enhancer elements in maize, wheat and other plants, we discussed the links between histone modifications, chromatin states, and open regions on TEs, and their effects on gene expression (paragraph 5 in “Discussion”)
In the newly added comparison and discussion above, we added nineteen citations of the references closely related to the topic, and we hope the revisions could greatly strengthen the depth and professionalism of our discussion.
Especially if the authors use in some graphic appendices a comparison of their results with, for example, Arabidopsis thaliana?
Response: Thanks for the reviewer’s good suggestion. The comparison of our results with other plants such as A. thaliana is a good idea. We have counted the content of TE and TE-related chromatin state of Arabidopsis, rice and maize, and found that Ae. tauschii and maize are very similar, with TE of the majority of the genome contributing to TE-related chromatin state, which is much higher than that of Arabidopsis and rice. We present the results of this comparison in Figure S7 and discussed them in the "Discussion" section. The data used for comparison were obtained from the PCSD database (http://systemsbiology.cau.edu.cn/chromstates/index.php).
Although it is stated in these figures that it is a comparison with A. thalina, the reader is not told where this data comes from, i.e. there is no adequate relevance of documenting the result for their comparison.
Response: We are sorry for not providing readers with the source of the data we use. The detail information about the antibodies we used was list in Table S1.We now have uploaded the ChIP-seq data captured with antibodies and the MNase-seq data used to obtain open regions used in our study to the National Genomics Data Center (NGDC, https://bigd.big.ac.cn). This data acquisition and accession number have been included in the “Data Availability Statement”.
The manuscript requires substantial and significant editing of the "Results" and "Discussion" sections.
Response: We appreciate very much for the reviewer’s suggestion. We have thoroughly revised the “Results” and “Discussion” sections of the manuscript accordingly. We substantially revised the “Results” section, and the main editing is as follows:
- All contents belonging to the discussion have been removed
- The basic description of these marks used in the chromatin state learning was added
- Added the assessment of the accuracy of chromatin open regions identification
- The summary of experimental results was added at the end of some paragraphs so that the mutually independent contents of the results are more coherent and consistent
- The causality in some conclusions were modified to be a correlation
- Some irregularities in our descriptions were revised
- We adjusted the citation of some figures and tables to make the correspondence between the results and citations more appropriate
Perhaps it would be appropriate to think about connecting the individual parts. It is through an adequate and critical discussion that authors can prove new knowledge in the given area.
Response: Thanks to the reviewer for the suggestion, this suggestion is very important. The reviewer gave us methodological guidance, which we greatly appreciate. Incomplete writing will hurt the presentation of our research and readers' ability to obtain useful information from our results. This suggestion is not only helpful for our writing of this article but also helpful for our future writing on how to properly discuss our results correctly.
At the same time, it would be appropriate to prepare a "Conclusion" section (it doesn't have to be long) with a summary of essential findings.
Response: We agree with the reviewer, we have added a short "Conclusion" section to the manuscript so that the results and discussion parts in the manuscript are summarized and condensed. We summarize our basic findings and possible contributions to academia through our conclusions.
Last but not least, we have improved the “Abstract” section to make the conclusions of the abstract closer to the conclusions of our article. Meanwhile, the revised manuscript has been edited by a professional editing service in the native English language to ensure that our results, discussions and conclusions are clearly presented to you and the readers.
Once again, we sincerely thank the reviewer for your time and comments. We hope to have addressed these concerns to the reviewer’s satisfaction, and any additional suggestions for improving our manuscript are welcomed.

Reviewer 2 Report
the manuscript is very interesting, the authors present interesting results. They are described in a very detailed way, additionally supported by numerous drawings and tables.
however, the work requires some corrections in particular regarding the discussion. in the discussion chapter, the authors cite only one publication, so they do not discuss their results against the background of others already published. the discussion in the manuscript is a summary of the results obtained, this should be changed.
moreover, they suggest in the description of the material to add information why this particular form was selected for analysis.
Author Response
Dear Reviewer,
Thank the reviewer for the comments concerning our manuscript entitled “Epigenetic landscape is largely shaped by diversiform transposons in Aegilops tauschii” (manuscript ID: ijms-2378947). Those insightful and constructive comments are all valuable and very helpful for revising and improving our manuscript, as well as the important guiding significance to our research. We have read the comments carefully and have made corrections which we hope meet with approval. The resubmitted manuscript is under revision mode and revised portions can be seen in our manuscript. The point-to-point responses to your comments are as follows:
the manuscript is very interesting, the authors present interesting results. They are described in a very detailed way, additionally supported by numerous drawings and tables. however, the work requires some corrections in particular regarding the discussion.
Response: Thank you very much for your affirmation of our work and for your suggestions. As you and another reviewer have commented, a lot of job need to do to improve our manuscript, especially the “Discussion” section. We have rewritten the “Discussion” section and compared our results with existing works on this topic according to your suggestions. We also revised the “Results” section accordingly. We hope the revised manuscript version will satisfy you.
in the discussion chapter, the authors cite only one publication, so they do not discuss their results against the background of others already published. the discussion in the manuscript is a summary of the results obtained, this should be changed.
Response: Thank you for the reviewer’s comments and constructive suggestions. We would like to apologize for our improper discussion, which was not adequately discussed against the background of others already published. This suggestion on how to organize discussion is very helpful for our writing of this article and our future research. Here, we did an in-depth discussion in the background of comparing with others. The main discussions added are as follows:
- In the context of wheat as an important staple crop and increasing environmental suitability of wheat, we discuss why tauschii is used, why TE and epigenetic landscape studies are conducted, and compared with studies on TEs in the Ae. tauschii genomic landscape (paragraph 1 in “Discussion”)
- In the context of the hypothesis of the “histone code”, we discuss why and how this method can establish the epigenetic landscape and further discussed the interesting results of TE states and bivalent states (paragraph 2 in “Discussion”)
- Compared with the chromatin state studies in mammals, Arabidopsis, rice, and wheat, we discuss the results of TE-related modifications in tauschii and the differentiation of chromatin states between TE superfamilies (paragraph 3 in “Discussion”)
- We discussed the possible reasons for hAT-Ac enrichment in open areas and compared them with a recent study on the prevalence and chromatin signatures of plant regulatory elements on TEs in plants (paragraph 4 in “Discussion”)
- Compared with the open features of promoter and enhancer elements in maize, wheat and other plants, we discussed the links between histone modifications, chromatin states, and open regions on TEs, and their effects on gene expression (paragraph 5 in “Discussion”)
In the discussion above, we added nineteen citations of the references closely related to the topic, and we hope the revisions could greatly strengthen the depth and professionalism of our discussion.
moreover, they suggest in the description of the material to add information why this particular form was selected for analysis.
Response: Thank you for the reviewer’s valuable suggestion. We agree with the reviewer. For this improvement, we have made the following revisions. We added the reason for selecting these marks for analysis in the first paragraph of the “Results” section. In the first paragraph of the “Discussion” section, we added the background of Ae. tauschii for analysis, and in the second paragraph, we added the background of the chromatin state method for analysis. In addition, we have added the background of AL8/78 as material in the “Materials and Methods” selection. Full disclosure of the material background and research background will help readers understand the reasons for our work and facilitate an understanding of our research results. We appreciate the reviewer very much for the constructive suggestion.
Thank you again for your time and valuable suggestions and apologies for our improper and inadequate discussion and material description. We hope to have addressed these concerns to the reviewer’s satisfaction, and any additional suggestions for improving our manuscript are welcomed. Meanwhile, the revised manuscript has been edited by a professional editing service in the native English language to ensure that our results, discussions and conclusions are clearly presented to you and the readers.

Round 2
Reviewer 1 Report
The authors made significant changes to the manuscript and answered most of the questions or reminders. From my point of view, the "Results" part still remains essential, where the general discussion is obvious in many places (e.g. line 80-82 or 113-117). Yes, I understand that links may appear in the results when using databases. In the case of some graphic supplements and comparing motifs, it would be good to indicate in the legend where the data from which they were created comes from, or to specify their number as well, thereby increasing the narrative ability and making the graphic attachments self-explanatory.
Author Response
Dear Reviewer,
We thank the reviewer for reviewing our revised manuscript and for your specific suggestions. Your comments and suggestions are so timely and constructive. Thank you very much for your careful and helpful work to improve our manuscript again.
We realize through your comments and suggestions that our previous revisions still have problems in the way the results are presented, the disclosure of the methods, and the self-explanatory of the charts. We have modified all the issues you mentioned. The point-to-point responses to your comments are as follows:
The authors made significant changes to the manuscript and answered most of the questions or reminders.
Response: We thank the reviewer for the approval of our efforts to improve the manuscript. This improvement is mainly due to your constructive comments and some good ideas from you and the other reviewer.
From my point of view, the "Results" part still remains essential, where the general discussion is obvious in many places (e.g. line 80-82 or 113-117)
Response: Thanks to the reviewer for the comment. We agree with you and we apologize for our omission. The result part is the core of an article, which is the most concerned place for readers, and also the place where the author should try to write well.
We have carefully examined all the paragraphs in the “Results” section and the corresponding charts, and found that there are still many contents belonging to the discussion. It is obvious that they should not be included in the “Results”. We have revised those contents, and have now moved them to the “Discussion” section or deleted them. You can view the revisions in Word revision mode.
We are so sorry that due to the revised manuscript we previously presented to you being in revision mode, there may be differences in the number of lines we see due to Word versions or Review modes. Therefore, we may have missed line 80-82 or 113-117 you mentioned. We guess that we may have found the place and have made revisions. If you find that that place have not been effectively revised in our current manuscript, please do not hesitate to remind us. We will make additional revisions as soon as possible.
We have now provided a PDF version in addition to the Word version, and I believe this mistake by us will not occur again.
Yes, I understand that links may appear in the results when using databases. In the case of some graphic supplements and comparing motifs, it would be good to indicate in the legend where the data from which they were created comes from, or to specify their number as well, thereby increasing the narrative ability and making the graphic attachments self-explanatory.
Response: We would like to apologize for that we did not clearly state the sources of the data that we used in our “Results” and attachments. This time, we directly cite the relevant papers in the figure legend instead of providing a link to the database. We added the data sources for comparing motifs in the supplemental figure legend. Furthermore, we have added the literature citations for all databases in the main text instead of offering just links. We also added a source or method description of the data for Arabidopsis, rice, and maize to the “Materials and Methods” section (section 4.8). Finally, we have examined and revised all the figures/tables and supplemental figures/tables to ensure the narrative ability of all the figures/tables for better presentation to readers.
Once again, we sincerely thank the reviewer for your time and comments. We hope to have addressed these concerns to the reviewer’s satisfaction, and any additional suggestions for improving our manuscript are welcomed. Thank you!
Round 3
Reviewer 1 Report
The authors accepted all my comments.